# The Sarcoplasmic Reticulum of Skeletal Muscle Cells: A Labyrinth of Membrane Contact Sites

**DOI:** 10.3390/biom12040488

**Published:** 2022-03-23

**Authors:** Daniela Rossi, Enrico Pierantozzi, David Osamwonuyi Amadsun, Sara Buonocore, Egidio Maria Rubino, Vincenzo Sorrentino

**Affiliations:** Department of Molecular and Developmental Medicine, University of Siena, Via Aldo Moro 2, 53100 Siena, Italy; pierantozzi@unisi.it (E.P.); davidosamwonuyi.a@student.unisi.it (D.O.A.); sara.buonocore@student.unisi.it (S.B.); egidiomaria.rubino@student.unisi.it (E.M.R.); vincenzo.sorrentino@unisi.it (V.S.)

**Keywords:** muscle, Ca^2+^, myopathy, intracellular membrane

## Abstract

The sarcoplasmic reticulum of skeletal muscle cells is a highly ordered structure consisting of an intricate network of tubules and cisternae specialized for regulating Ca^2+^ homeostasis in the context of muscle contraction. The sarcoplasmic reticulum contains several proteins, some of which support Ca^2+^ storage and release, while others regulate the formation and maintenance of this highly convoluted organelle and mediate the interaction with other components of the muscle fiber. In this review, some of the main issues concerning the biology of the sarcoplasmic reticulum will be described and discussed; particular attention will be addressed to the structure and function of the two domains of the sarcoplasmic reticulum supporting the excitation–contraction coupling and Ca^2+^-uptake mechanisms.

## 1. Introduction

The sarcoplasmic reticulum (SR) is a specialized form of the endoplasmic reticulum of muscle cells, dedicated to calcium ion (Ca^2+^) handling, necessary for muscle contraction and relaxation. Studies with electron microscopes (EMs) have revealed that, in striated muscle cells, the SR is organized into numerous interconnected tubules forming a network of longitudinal elements surrounding each myofibril [1]. These elongated tubules, known as longitudinal SR (l-SR), are dedicated to the removal of Ca^2+^ from the cytosol, an activity operated by the sarcoplasmic/endoplasmic reticulum Ca^2+^ ATPases (SERCA) that actively pump Ca^2+^ from the cytoplasm to the lumen of the SR. In skeletal muscle fibers, the l-SR is localized around the A and I bands of each sarcomere. At regular intervals, which correspond to the borders between A and I bands of sarcomeres, the tubules of the l-SR merge into enlarged sac-shaped structures, known as terminal cisternae. Here, two terminal cisternae are positioned at the opposite sides of tubular infoldings in direct continuity with the plasma membrane, named transverse tubules (TT) (Figure 1). The structure formed by two terminal cisternae and one TT, called a "triad", represents the membrane platform where several dedicated proteins operate in transducing the depolarization of the plasma membrane into Ca^2+^ release from the SR, a mechanism known as excitation–contraction coupling (ECC). The region of the terminal cisternae that opposes the TT membrane is called junctional SR (j-SR), and, in this region, the Ryanodine Receptors type 1 (RyR1) Ca^2+^ release channels are localized.

### sAnk1.5 and Obscurin Stabilize the SR around the Myofibrils

The proper relationship between the elongated tubules of the l-SR and myofibrils is a pivotal structural prerequisite to organize the different domains of the SR and allow an efficient muscle contraction [1,2]. As of today, the only two known proteins responsible for the correct SR juxtaposition and stabilization around myofibrils are the small muscle-specific isoform of ankyrin 1 (sAnk1.5) localized on the SR membrane, and obscurin, a giant sarcomeric protein [3,4,5,6]. sAnk1.5 is a striated muscle-specific isoform encoded by the ANK1 gene, containing a transmembrane domain that allows its insertion in the SR membrane, in correspondence with the M bands and, to a lesser extent, the Z disks. The C-terminal cytosolic tail of sAnk1.5 contains an ankyrin-like repeat that mediates binding to the non-modular region at the COOH-terminal of obscurin [3,4,5,6]. Similar to sAnk1.5, obscurin shows localization at the M band and Z disk of the sarcomere [6,7,8,9,10,11]. Several pieces of evidence obtained in mice knockout for either obscurin or sAnk1.5 show a significant reduction in the volume of l-SR tubules around the myofibrils even when only one of the two proteins is absent, confirming that the interaction between sAnk1.5 and obscurin is crucial to stabilize the SR around the myofibrils [7,8].

## 2. The Triad, a Unique Membrane System of the Skeletal Muscle

The triad is the membrane structure that supports ECC in skeletal muscle fibers by allowing the interaction between the dihydropyridine receptor (DHPR) and RyR1 [12]. The DHPR is a voltage-dependent L-type Ca^2+^ channel located on the TT that, following sarcolemma depolarization induced by motor-neuron stimulation, undergoes a conformational change that allows the opening of RyR1, the Ca^2+^ channel located on the terminal cisternae of the j-SR, resulting in a massive Ca^2+^ efflux from the SR into the myoplasm and activation of muscle contraction [13].

### Triad Biogenesis, Repair, and Maintenance

The events that characterize TT maturation and triads formation have been morphologically defined by electron and confocal microscopy studies. These revealed that the juxtaposition of j-SR and TT begins in the embryonic (E) period of development and evolves through sequential steps in the postnatal life [14,15]. In mammals, the first step in triads biogenesis is represented by the formation of a tubular SR adjacent to myofibrils, which will subsequently develop into reticular structures surrounding the myofibrils [16]. In mouse embryos, a detectable SR is observed as early as day E14; by day E15, immunohistochemical analysis reveals the presence of RyR1 clusters positioned at the periphery of developing myotubes, and, by day E16, RyR1 clusters start aligning with the A-I borders of the sarcomeres. On day E17 and continuing after birth, the j-SR progressively acquires a transverse orientation, with RyR1 showing its typical double-row distribution [14,15].

TT development appears to start immediately after that of the SR. At day E15, small tubular invaginations can be observed at the periphery of developing muscle fibers. By day E19, these invaginations expand deeper into the fiber, although remaining almost longitudinally oriented. After birth, the TT system starts acquiring its final transverse position over the borders between the A and I bands, achieving its definitive localization at around three weeks of age in mice [2,14]. A recent study performed in zebrafish that combined live imaging with three-dimensional electron microscopy proposes that membrane-derived tubules start forming at transient sarcolemma endocytic nucleation sites. These structures further grow by membrane traffic-based mechanisms and interact with the sarcomere, likely via the SR, finally resulting in an ordered transverse organization [17]. The association between SR and sarcolemma membranes is already evident as early as E14–E15, mainly at the periphery of the fiber. Starting on E16, internal junction sites appear as diads, formed by one developing TT flanked by a single terminal cisterna. On days E17–E18, most SR/TT junctions are structured as triads [18]. During in vitro differentiation of skeletal muscle, myoblasts recapitulate the main steps of in vivo SR and TT development, as reported in Figure 2.

While several details of the morphological events that characterize TT and SR maturation have been defined, the molecular mechanisms that drive this process remain more elusive. RyR1 and DHPR, although essential for ECC, have been shown to be dispensable for the formation of triads; indeed, triad formation was observed in knockout mice for RyR1, also called “dyspedic” mice since RyRs were defined as “junctional feet”. Triad formation was also observed in mice carrying muscular dysgenesis in DHPR, a natural mutation that causes lethal paralysis in mice. Finally, mice lacking both RyR1 and DHPR also show triad formation, further supporting the idea that these two proteins are not directly involved in triad formation or maintenance [19,20,21]. Several lipid-binding proteins and enzymes regulating phospholipid metabolism have been proposed to participate in TT and j-SR biogenesis and maintenance, such as amphiphysin 2, Caveolin-3, Dysferlin, Mitsugumins, Myotubularin, and Junctophilins [22] (Figure 3).

Amphiphysin 2, also known as Bridging integrator-1 (Bin1), is a member of a family of ubiquitous proteins able to shape lipid membranes; thanks to their BAR (Bin/Amphiphysin/Rvs) domain, they can interact with membrane lipids and detect and/or generate membrane curvatures [24]. Bin1 can drive membrane infoldings, thus supporting the generation of TT in muscle cells [25]. In skeletal muscle cell differentiation, Bin1 shuttles between the nucleus and cytoplasm [26] and can recruit and partially inhibit the embryonic isoform of Dynamin2 (DNM2), a GTPase involved in membrane fission and endocytosis [27,28]. Interestingly, Bin1 is not able to affect the activity of the adult isoform of DNM2, suggesting that it supports the first steps of membrane tubulation while, in the adult, it contributes to stabilizing the TT system [27]. The concurrent role of Bin1 and DNM2 in TT biogenesis has been confirmed by studies performed in Bin1 knockout mice [29] and by the identification of mutations in both genes that are causative of human centronuclear myopathy [30,31]. The SH3 domain of Bin1 also forms a complex with N-WASP that regulates the assembly of the actin cytoskeleton required for structural stability and organization of triads. This complex appears to include gamma-actin and tropomyosin Tm5NM1 in the proximity of the triads [32,33].

Caveolin-3 (Cav-3) is the muscle-specific isoform of caveolins, which mediates the formation of peripheral invaginations in cholesterol-rich regions of the plasma membrane [27,34,35]. Cav-3 knockout mice develop morphological abnormalities in the TT system consisting of tubule dilation and loss of transversal orientation [36]. In the adult, caveolae concentrate at sites of membrane injury where the resorption of damaged material is mediated through caveolae-dependent endocytosis [37]. The process of caveolae-mediated repair is inhibited or impaired in caveolinopathies, including limb–girdle muscular dystrophy (LGMD1C) [38], rippling muscle disease [39], and distal myopathy [40], all characterized by TT structural alterations. In this context, a key role is also played by another protein family, the cavins, which have been described as essential structural components of caveolae [41]. Cavin-4 is expressed predominantly in muscle, and its distribution is perturbed in human muscle diseases associated with Cav-3 dysfunction [42]. Cavin-1 (also known as Polymerase I and transcript release factor) is also expressed in skeletal muscle, where it works as a docking protein, together with Mitsugumin 53, during acute cell damage and membrane repair [43]. In addition to cavins, Cav-3 also interacts with Dysferlin and Mitsugumin 53 during membrane repair [44,45,46].

Dystrophy-associated fer-1-like protein (DYSF) is an integral protein belonging to the ferlin family, characterized by multiple C2 domains able to bind membrane phospholipids in the presence of Ca^2+^ [47]. It is mainly involved in membrane repair. Indeed, DYSF-deficient myofibers show a slower sarcolemma resealing and contain more dilated, irregularly shaped, and longitudinally oriented TT, with abnormal accumulation of vesicles at damaged sarcolemma sites [48,49]. Mutations in the human *DYSF* gene cause myopathies, including limb–girdle muscular dystrophy type 2B (LGMD2B), Miyoshi myopathy (MM), and distal anterior compartment myopathy [50,51,52]. DYSF, together with the membrane-associated actin-binding proteins annexin A6, annexin A2a, and annexin A1a, is concentrated in the vicinity of membrane injury, where it regulates Ca^2+^-dependent vesicle fusion with the plasma membrane [53]. Annexins are also recruited at the sarcolemma by Anoctamin 5 (ANO5/TMEM16E), a Ca^2+^-activated chloride channel, which was also suggested to display phospholipid scramblase activity [54]. Mutations in ANO5/TMEM16E are associated with autosomal recessive limb–girdle muscular dystrophy-12 [55,56]. More recently, Chandra and co-workers proposed that ANO5/TMEM16E may play a dual role in membrane repair since, in addition to recruiting annexins to the injury sites, it also mediates counter anion influx into the SR, supporting the activity of SERCA pumps in clearing the cytoplasm from Ca^2+^ overload due to membrane damage [57].

Mitsugumin-53 (MG53), also known as tripartite motif 72 (TRIM72), is a skeletal and cardiac muscle-specific protein that contributes to vesicle trafficking involved in the membrane repair machinery [46,58]. MG53 expression increases during myogenesis, and, during membrane damage repair, it is recruited at damaged membrane sites by cavin-1 where, by binding to phosphatidylserine, it acts as a scaffold to recruit additional proteins, such as Cav-3, DYSF, and annexin V, to start vesicle trafficking [49,59,60]. Indeed, MG53 knockout mice develop progressive muscle pathologies, likely correlated with an impediment in the membrane repair mechanism [49]. Surprisingly, no MG53 mutation associated with the onset of skeletal muscle pathologies has been described so far.

Mitsugumin-29 (MG29), which belongs to the synaptophysin family and is characterized by early expression during myogenesis, associates with newly formed SR vesicles [61,62]. During triad maturation, MG29 can be detected on both the SR and TT, while, in adult skeletal muscle, it localizes exclusively at triads [63]. MG29 knockout mice are characterized by decreased muscle mass and overall muscle weakness, likely due to the presence of swollen TT and less organized triads [64,65]. In vitro studies on mouse primary skeletal myotubes also showed that MG29 interacts with the canonical-type transient receptor potential cation channel 3 (TRPC3), localized at the TT, suggesting the existence of an additional mechanism able to regulate Ca^2+^ transients during skeletal muscle contraction [66].

Myotubularin (MTM1) is a lipid phosphatase localized at triads. Mutations in human *MTM1* are associated with X-linked myotubular myopathy, a skeletal muscle disorder that results in severe muscle weakness with abnormal nuclei positioning [67]. Mice knockout for MTM1 present with progressive centronuclear myopathy characterized by SR and TT disorganization, centrally positioned nuclei, and disorganized mitochondria distribution. Interestingly, these structural alterations are more evident with aging, suggesting that MTM1 might play a role in later stages of TT formation and/or in maintaining the stability of TT, rather than in the early biogenesis of triads [68]. MTM1 dephosphorylates phosphatidyl-inositol-3-phosphate (PtdIns3P) and phosphatidyl-inositol-3,5-bisphosphate (PtdIns3,5P2), two second messengers that regulate docking and fusion of vesicles with the plasma membrane [69]. This suggests that muscle damage observed in MTM1 knockout mice might be due to abnormal endosomal vesicle trafficking and autophagic fluxes [70]. On the other hand, overexpression of MTM1 also led to the formation of abnormal membrane structures that stained positive for Cav-3, dystrophin, and DHPR [71], indicating that balanced expression levels of MTM1 are required for the proper assembly of SR and TT into triads. Interestingly, MTM1 co-localizes and interacts with BIN1 and can enhance BIN1-mediated membrane tubulation, indicating that MTM1 may also be involved in regulating membrane curvature by acting on PtdIns3P-rich membrane subdomains [72,73]. Finally, MTM1 was found to interact with the striated muscle enriched protein kinase SPEG [74]; mutations in *SPEG* have been linked to centronuclear myopathies with disruptions of triadic structures and impairment of ECC [75].

Junctophilins mediate the apposition between ER/SR and plasma membrane in different cell types. In mammals, Junctophilin 1 (JPH1) and Junctophilin 2 (JPH2) are expressed in striated muscles [76,77,78]. All isoforms share a similar protein structure consisting of two groups of membrane occupation and recognition nexus (MORN) motifs, separated by a joining region followed by an α-helix, a divergent region, and a transmembrane domain at their C-terminus [79]. MORN motifs can recognize and interact with membrane phospholipids [76,80]. The α-helical region is highly conserved and provides mechanical elasticity to the protein, while the divergent region, which displays the lowest similarity among the four isoforms, still lacks a clear functional role [80]. The short transmembrane domain anchors JPHs to the ER/SR membrane and, together with MORN motifs—which interact with plasma membrane phospholipids—allows JPHs to establish a stable contact site keeping the two cell membranes 12–15 nm apart [77,80].

In skeletal muscle, both JPH1 and JPH2 are present at triads, where they are essential for the formation and maintenance of these contact sites. In addition to their structural role, JPHs can also bind and regulate various proteins at triads, acting as a scaffold for assembly of the Ca^2+^ release complex [81,82,83]. JPH1 and JPH2 form homo- and heterodimers that can interact with several proteins on the SR and the sarcolemma, including RyRs, DHPR, Cav-3, and the TRPC3 channel [84,85,86,87,88,89,90,91]. JPHs are also required to support the organization of the Store Operated Calcium Entry (SOCE) pathway [92]. Downregulation of JPHs results in the defective formation of triads and diads and altered Ca^2+^ homeostasis due to a partial loss in RyR1 and DHPR co-localization [84,85,86]. JPH1 knockout mice die shortly after birth, presenting triad alterations and impaired ECC [93,94], while JPH2 knockout mice show embryonic mortality due to significant alterations in diads and cardiac ECC [77]. In the heart, remodeling and loss of TT in chronic heart failure, in aging or in cardiomyopathies correlate with downregulation of JPH2 protein expression and/or loss of JPH2 localization at TT; in addition, mutations in JPH2 are linked to the development of cardiomyopathies [95,96,97,98,99,100]. Two mechanisms have been suggested to explain JPH2 downregulation in cardiac diseases. The first is associated with the up-regulation of microRNA-24 (miR-24) observed in failing hearts; this microRNA can bind the 3′ untranslated region of JPH2 mRNA, resulting in the downregulation of JPH2 translation [95]. The second mechanism is due to calpain-mediated protein cleavage [95]; several studies showed that both JPH1 and JPH2 are targets of the calcium-depended protease μ-calpain (or calpain-1) and m-calpain (or calpain-2), whose activity increases in response to stress and/or unbalanced Ca^2+^ homeostasis [101,102,103,104,105]. In cardiac muscle, calpain-dependent proteolysis may explain the downregulation of JPH2 observed in heart failure [103,104]. Nevertheless, while JPH degradation results in the modification of the geometry of triads and diads, leading to ECC alterations, the N-terminal fragment of JPH2 generated by calpain digestion in stressed cardiac muscle was found to translocate to the cell nucleus, where it acts as a transcriptional regulator mitigating the hypertrophic response to cardiac disease [101,102,103,104,105].

## 3. The Protein Complex of the ECC

At triads, several proteins participate in the ECC process, including integral membrane proteins such as RyR1, DHPR, triadin, junctin, j-SR protein 1 (jp45), and mitsugumin-56, the STAC3 adaptor protein, or luminal proteins such as the Ca^2+^ binding proteins calsequestrin (CASQ) and histidine-rich calcium (HRC) binding protein [14,23,106,107] (Figure 3).

### 3.1. RyR1, DHPR, and STAC3 Are Essential for ECC

RyRs are a family of Ca^2+^ release channels that represent the largest ion channels known to date. In skeletal muscle, two RyR isoforms, RyR1 and RyR3, are expressed, although only RyR1 is essential for ECC activation. In skeletal muscle, RyR1 is primarily activated by a conformational coupling with the DHPR, a mechanism defined as depolarization-induced Ca^2+^ release (DICR) [13]. At variance with RyR1 channels, RyR3 channels are mainly expressed in neonatal versus adult muscles and are activated by a Ca^2+^-induced Ca^2+^-release (CICR) mechanism, whereby an increase in cytosolic Ca^2+^ causes RyR3 opening and, thus, Ca^2+^ release from intracellular stores [13,108,109]. RyR2 channels are mainly expressed in cardiac muscle, where they are involved in cardiac ECC through a CICR mechanism [106]. RyRs have a tetrameric structure, made by the assembly of four identical monomers, each consisting of a large N-terminal region of about 4300 amino acids extending in the sarcoplasm, while the remaining region contains the six transmembrane helices that anchor each monomer to the SR membrane and contribute to forming the channel pore region, followed by a short cytoplasmic C-terminal tail [110,111]. EM reconstruction studies have shown that RyRs display a mushroom-like form, with the cap in the cytoplasm representing 80% of the volume, and the stalk anchored in the j-SR membrane. In the cytoplasmic region, many domains have been described, referred to as subregions, that represent binding sites for several auxiliary proteins and molecules that contribute to regulating the opening and closing of the channel [110,111,112]. Calmodulin (CaM) was proposed to bind at different regions of RyR1 channels, acting either as a weak activator, at nanomolar Ca^2+^ concentration, or, in the micromolar range, as a channel inhibitor [113]. One of the proposed binding sites for CaM also binds the EF-hand protein S100, a protein also able to activate the channel [114]. RyR1 is also regulated by a member of the FK506-binding protein family, namely, FKBP12, which stabilizes the channel in a closed state [110,111,115,116]. Additional physiological or pharmacological RyR1 regulators include Mg^2+^, Homer 1c, the scorpion venom imperatoxin, halothane, dantrolene, PCB95, ryanodine, ruthenium red, methylxanthines, and 4-chloro-m-cresol [111,117,118]. Among these, dantrolene deserves special attention since it is now the only available agent for the treatment of malignant hyperthermia (MH), a pharmacogenetic disorder that results in a hypermetabolic state following exposure to volatile anesthetics or succinylcholine. MH is mainly associated with dominant mutations in human *RYR1* that result in RyR1 hyperactivation, leading to massive and uncontrolled Ca^2+^ efflux from the SR. Dantrolene was historically used as a skeletal muscle relaxant, and later, it was demonstrated to be able to block calcium release from the SR. Although the molecular mechanism responsible for dantrolene action on RYR1 is still not completely defined, its introduction for the clinical treatment of MH led to a significant decrease in mortality, from more than 90% to less than 5% [119]. Finally, RyRs can also be regulated by post-translational modification, such as oxidation, nitrosylation, and phosphorylation/dephosphorylation cycles that occur via several kinases, such as PKA, PKG, Ca^2+^/CaM-dependent protein kinase II (CaMKII), and phosphatases (PP1, PP2A, and PDE4D3) [111,120].

DHPRs are L-type voltage-gated Ca^2+^ channels located on the TT. The skeletal muscle DHPR is composed of a heteromultimeric complex that includes the *α*1_s_, α2, and δ, β1α, and γ1 subunits [121]. The *α*1_s_ subunit (also referred to as Ca_V_1.1) is an integral membrane protein containing four transmembrane domains, each composed of six alpha helices, acting as the pore-forming and the voltage-sensing unit. The voltage-sensing domain of *α*1_s_ interacts with the γ1 subunit that regulates channel inactivation [122]. The *β*1*_a_* subunit is also located in proximity to the voltage-sensing domain of the *α*1_s_ subunit and supports the assembly of DHPRs in tetrads, where one DHPR tetrad faces one of the four subunits of the tetrameric RyR1 channel [121,123,124]. Mice knockout for either *α*1_s_ or *β*1*_a_* subunits die perinatally, indicating that these subunits are essential for ECC [125,126]. The α2-δ subunits associate through four disulfide bonds and interact with the extracellular loops of the *α*1_s_ subunit [121]. Despite its function as a voltage-gated calcium channel, the role of DHPR in skeletal muscle ECC is to induce Ca^2+^ release from the SR by physically interacting with RyR1, according to “orthograde” signaling [127]. RyR1 is also able to regulate DHPR activity by the so-called "retrograde" coupling effect [17,128]. In particular, it was demonstrated that, in the absence of RyR1, the Ca^2+^ current mediated by DHPR is smaller than that measured in the presence of RyR1, suggesting that RyR1 can promote the Ca^2+^ channel activity of DHPR and accelerates DHPR activation. Interestingly, this activity is unique to RyR1 since RyR2 was found not to be able to restore either orthograde or retrograde signaling. Different regions in DHPR and RyR1 have been proposed to be important for their reciprocal association and regulation; the II-III loop of the *α*1_s_ subunit is generally accepted to be involved in both orthograde and retrograde coupling, while the site of interaction on RyR1 has not yet been completely defined [14,127,129].

STAC3 is a recently identified component of the ECC machinery. It is an adaptor protein that supports the trafficking of the *α*1_s_ subunit of DHPR and regulates the coupling of DHPR with RyR1 [129,130,131,132]. STAC3 knockout mice present severe defects in muscle development, mass, and morphology; they are completely paralyzed and die shortly after birth [132]. Mutations in *RYR1, DHPR,* and *STAC3* were identified in several human congenital myopathies, as summarized in Table 1 [119,133,134,135,136].

### 3.2. Triadin and Junctin (JNT), Two Integral Membrane Proteins That Regulate ECC

Triadin is encoded by the TRDN gene that generates, via alternative splicing, four different isoforms named according to their molecular weight: the skeletal muscle isoforms are Trisk 95 and Trisk 51, both displaying a triadic localization; the main cardiac isoform Trisk 32 (also known as CT1) and Trisk 49, which does not localize at the j-SR in skeletal muscle fibers [137,138]. Triadin isoforms share a common cytoplasmic N-terminal and transmembrane domain but differ in the length and composition of their luminal C-terminal segment [139]. Trisk 95 contains two cysteine residues in its luminal domain, C270 and C649, that enable self-multimerization and is targeted at triads by specific domains localized both in the cytoplasmic and the luminal regions [140,141]. Triadin acts as a functional regulator of ECC by interacting with RyRs [142,143], junctin [144,145], calsequestrin [141,146,147,148], and the histidine-rich Ca^2+^-binding protein (HRC) [149]. Triadin may also play a structural role in supporting triad architecture by interacting with the microtubule-binding protein Climp-63, also known as Cytoskeleton-associated protein 4 (CKAP4) [150]. Triadin knockout mice present both ECC alterations and abnormal triads [151,152]. In contrast to heart muscle, where mutations in triadin are associated with heart disease, no skeletal muscle disease has been associated with triadin so far.

Junctin is structurally similar to triadin and, like triadin, can bind calsequestrin and the ryanodine receptor [144,145,153,154,155]. Junctin knockout mice exhibited increased contractility and Ca^2+^-cycling parameters in cardiac muscle [156], but no significant changes were observed in skeletal muscle [157]. Similarly, acute junctin downregulation in cardiomyocytes resulted in a significant increase in contraction [158], while in skeletal myotubes, it resulted in a decrease in Ca^2+^ release, suggesting that junctin may display a different role in cardiac and skeletal muscles [159].

### 3.3. Calsequestrin- and Histidine-Rich Calcium-Binding Protein Store Ca^2+^ in the SR Lumen

In skeletal muscle, two isoforms of calsequestrin (CASQ) have been identified: CASQ1 and CASQ2. CASQ1 is expressed in fast- and slow-twitch skeletal muscle fibers, whereas CASQ2 is expressed in slow-twitch skeletal muscle fibers and in cardiac muscle [160]. The two isoforms present a high sequence homology and basically only differ in their acidic C-terminus [161]. CASQs are intra-luminal SR soluble proteins with high capacity and low-affinity Ca^2+^-binding properties. [162,163]. CASQ1 exists as either a monomer or polymers, whose assembly depends on the SR Ca^2+^ concentration [164,165]. CASQ1 is formed by three negative thioredoxin-like domains surrounding a hydrophilic core [166]. Park and collaborators proposed a model where, when the ionic strength in the lumen of the SR increases, these domains fold together, and CASQ monomers start to polymerize. Following a further increase in Ca^2+^ concentration, front-to-front and back-to-back interactions take place. This finally results in the assembly of large ribbon-like structures, where negatively charged cavities may accommodate additional Ca^2+^ [167]. CASQ1 polymerization is a reversible process depending on the luminal Ca^2+^ concentration and the presence of CASQ-binding proteins or post-translational modification [168,169,170,171]. During sustained SR Ca^2+^ depletion, it was shown that almost all CASQ is present in its depolymerized status. In this condition, progressive closure of the RyRs was reported, with a consequent reduction in Ca^2+^ release aimed to prevent dangerous levels of SR Ca^2+^ depletion. Accordingly, it has been proposed that CASQ1 depolymerization may represent the intracellular switch that induces RyR1 closing [172].

In skeletal muscle, CASQs interact with RyR1, triadin, and junctin, forming a quaternary Ca^2+^ release complex [144,145,148,173,174]. However, how these interactions regulate ECC still remains to be defined, since different studies report that CASQs may either inhibit or activate or even have no effect on the opening of RyRs [169,174,175,176,177,178,179,180,181]. CASQ1 knockout mice show mild atrophy, narrower terminal cisternae, and proliferation of multilayered junctions, as well as mitochondria alterations. They exhibit significantly reduced SR Ca^2+^ content and marked SR Ca^2+^ depletion during high-frequency stimulation [182,183]. Recently, it has been demonstrated that CASQ1 acts as a regulator of SOCE by binding to STIM1. In particular, CASQ1 overexpression was found to inhibit SOCE by reducing STIM1 clustering and STIM1/OraiI interaction [184,185]. Indeed, in CASQ1 knockout mice, SOCE is constitutively active, resulting in enhanced Ca^2+^ entry that may compensate for the reduced total SR Ca^2+^ content [186,187]. Mutations in the *CASQ1* gene have been identified in patients affected by a rare vacuolar myopathy [188] and in patients with tubular Aggregate Myopathy, TAM [189,190].

HRC is far less abundant than CASQs, suggesting that it works as a secondary calcium-binding protein [191,192]. It is composed of a conserved N-terminal domain, a C-terminal cysteine-rich region, and a central histidine-rich region, which allow HRC to bind Ca^2+^ and to interact with triadin [193,194]. Under physiological Ca^2+^ levels, HRC has a pentameric structure, but when Ca^2+^-binding sites are saturated due to increased SR Ca^2+^ concentration, the protein shifts to a monomeric form [195]. Although considered as a secondary calcium-binding protein, HRC appears to play a non-secondary role in regulating Ca^2+^ homeostasis. Indeed, HRC may function as a negative regulator of RyRs and, in cardiac muscle, it was suggested to interact with SERCA pumps, acting as a negative regulator of Ca^2+^ re-uptake [149,192,196,197].

## 4. SERCA Pumps at the l-SR Are Responsible for Ca^2+^ Re-Uptake from the Sarcoplasm

The l-SR is mainly involved in Ca^2+^ re-uptake, although some ligand-gated Ca^2+^ channels, such as inositol 1,4,5-trisphosphate receptors (InsP_3_R), which release Ca^2+^ to regulate signaling pathways apart from myofibrils contraction, are also present here [198]. SERCA pumps play the most important role in SR Ca^2+^ replenishing. SERCAs belong to the P-type ATPases family, a large superfamily of integral membrane proteins that pump ions and lipids across cellular membranes using the energy derived from ATP hydrolysis. The P-type ATPases family is composed of five subfamilies, defined as P1–P5. SERCAs belong to the P2A-ATPases subfamily and are specific for Ca^2+^ transport, while P4-ATPases act as phospholipid flippases to maintain lipid asymmetry in a variety of membranes [199,200]. SERCA pumps can transport two Ca^2+^ ions for each hydrolyzed ATP. In vertebrates, SERCA pumps are encoded by three different genes, named ATP2A1-3. More than 10 protein variants are generated through alternative splicing occurring in the 3′-end of the main transcripts [201,202].

*ATP2A1* encodes two major skeletal muscle proteins: SERCA1a and SERCA1b, highly expressed in fast-twitch skeletal muscle fibers. SERCA1b is expressed during neonatal stages and in regenerating muscles, while it is replaced by SERCA1a in adult skeletal muscles [203,204,205]. Four isoforms of SERCA2 (a–d) are generated by alternative splicing of the *ATP2A2* gene. SERCA 2a is expressed in cardiac muscle, slow-twitch skeletal muscle fibers, and smooth muscle cells. SERCA2b is reported to be ubiquitously expressed, and it represents the main SERCA isoform in the brain [206,207,208]. mRNA of SERCA2c was found in epithelial, mesenchymal, and hematopoietic cell lines, as well as in primary human monocytes [206] and cardiac muscle, while SERCA 2d has been identified in skeletal muscle, although its role has not yet been fully elucidated [209]. *ATP2A3* codes for six isoforms (SERCA3a–f), mostly identified in non-muscle cells, although SERCA3a, 3d, and 3f were also detected in cardiac muscle cells [209,210]. The primary structure of the different SERCA isoforms is highly conserved. Nevertheless, the SERCA variants differ in their enzymatic properties. For instance, SERCA1a displays a maximal activity that is two-fold higher than that of SERCA2a [211,212], and differences in Ca^2+^ affinities have also been reported between SERCA2b and SERCA3 [212]. SERCA activity is finely regulated by numerous small proteins. The first identified regulator of SERCA is Phospholamban (PLN), which exerts an inhibitory effect on SERCA in cardiac muscle [213,214,215] and, to a lesser extent, in smooth muscle and slow-twitch oxidative skeletal muscle fibers where, however, its functional role appears to be marginal [216,217,218]. PLN exists in two forms, monomeric and pentameric. The first is considered the “active” inhibitory form, while the pentameric state is considered the “inactive” storage form of PLN. PLN regulation occurs through one-to-one interaction with SERCA [218]. At high cytosolic Ca^2+^ concentrations, the inhibitory effect of PLN is mitigated due to Ca^2+^/calmodulin kinase (CaMKII)- and/or Protein Kinase A (PKA)-dependent phosphorylation [219,220,221]. A second inhibitor of SERCA in skeletal muscle and atria is represented by sarcolipin (SLN). In contrast to PLN, which inhibits SERCA activity by reducing its affinity for Ca^2+^, SLN uncouples ATP hydrolysis from Ca^2+^ transport [222], thus promoting futile cycling, resulting in ATP consumption and increased thermogenesis. Indeed, SLN knockout mice do not show any visible phenotype but show compromised thermogenic capacity when exposed to cold [222,223]. This suggests that the role of SLN may be related to muscle functions linked to body metabolism and heat production rather than contraction [223]. Similar to the j-SR, the l-SR also contains a Ca^2+^ binding protein, namely, Sarcalumenin (SAR) [224], which colocalizes and interacts with SERCA2 [225]. Furthermore, SAR has been shown to play an essential role in maintaining cardiac function under biomechanical stresses and endurance exercise training [226,227]. Mutations in SERCA1 are associated with Brody disease, an extremely rare autosomal recessive myopathy (with a prevalence estimated at around 1 in 10 million births) characterized by muscle stiffness, myalgia and muscle cramps, aggravation of symptoms upon exposure to cold temperatures, and, in a smaller percentage of patients, muscle weakness. It is basically characterized by exercise-induced impairment of fast-twitch skeletal muscle relaxation due to diminished SERCA1 activity. This increased levels of cytosolic Ca^2+^ due to SERCA1 impairment directly contribute to muscle stiffness [228,229,230].

Although SERCAs represent the key enzymes that contribute to sarcoplasmic Ca^2+^ clearance following muscle contraction, a trans-sarcolemmal flux toward the extracellular environment also occurs in striated muscles. This is mediated by Plasma Membrane Calcium ATPases (PMCAs) and Na^+^/Ca^2+^ exchangers localized in the sarcolemma. Similar to SERCAs, PMCAs also belong to the P-type ATPase superfamily. They can transport one Ca^2+^ for each hydrolyzed ATP and are directly activated by interaction with Ca^2+^-calmodulin, which enhances PMCAs affinity for Ca^2+^ [231]. PMCAs are encoded by four different genes (ATP2B1-4) that, by alternative splicing, generate multiple isoforms [232]. Among these, PMCA3 and PMCA1 were found to be expressed in skeletal muscle, and PMCA1 was specifically localized on the T-tubule membrane [233,234]. NCX catalyzes the exchange of three Na^+^ ions and one Ca^2+^ ion across the sarcolemma in a high capacity and low Ca^2+^ affinity manner. Depending on the electrochemical gradient across the plasma membrane, NCX can either extrude intracellular Ca^2+^ (according to the so-called forward mechanism) or take up extracellular Ca^2+^ (a reverse mechanism) [235]. In mammals, three genes, NCX1, NCX2, and NCX3, are present; NCX3 is expressed in brain and adult skeletal muscle [236,237], while NCX1 is expressed in developing skeletal muscles and in cardiac muscle [237]. The role of NCX has been mainly defined in cardiac muscle cells, where it represents the major mechanism of Ca^2+^ extrusion from the cytoplasm during diastole [238,239].

## 5. Ca^2+^ Entry Units (CEU): Novel SR/Plasma Membrane Contact Sites to Refill Intracellular Ca^2+^ Stores

Entry of Ca^2+^ from the extracellular environment in response to a decrease in the endoplasmic reticulum Ca^2+^ content is a ubiquitous mechanism present in all cell types. The key elements of this pathway, named Store Operated Calcium Entry (SOCE), are STIM1 and Orai1 proteins [240]. STIM1 is a transmembrane protein localized in the ER membrane that acts as a sensor of luminal Ca^2+^ content; when intracellular Ca^2+^ stores are full, it is in a Ca^2+^-bound monomeric conformation. Depletion of the intracellular Ca^2+^ store results in dissociation of Ca^2+^ from STIM1, which forms aggregates that relocate at ER–plasma membrane contact sites, where they interact with and activate the Orai1 channels on the plasma membrane, allowing Ca^2+^ entry from the extracellular environment and refill of intracellular stores. Ca^2+^ entry from the extracellular environment through the SOCE mechanism is also present in skeletal muscle [241]. More recently, the requirement of extracellular Ca^2+^ to refill the SR Ca^2+^ stores, after intense, prolonged activity, has been shown to result in the assembly of additional intracellular junctions between the SR and the plasma membrane, providing additional space for STIM1 and Orai1 interaction. These newly identified sites, named Ca^2+^ Entry Units (CEU), support increased Ca^2+^ entry via Orai1, and their activity appears to contribute to improving fatigue resistance under continuous muscle activity [242,243,244] (Figure 4). Mutations in Orai1 and STIM1 have been identified in patients with tubular aggregate myopathy (TAM), a rare genetic disease characterized by accumulation, in type-2 fibers, of highly ordered membrane tubules containing SR proteins, such as CASQ1, SERCA, triadin, RyR1, and STIM1 [245,246,247,248,249]. More recently, mutations in CASQ1 have also been identified in TAM patients [189,190].

## 6. Mitochondria-Associated Membranes (MAM)

Interaction between the ER/SR and mitochondria occurs at specific membrane contact sites, generally defined as Mitochondria Associated Membranes (MAMs), populated by several proteins and protein complexes that act as tethers between the SR and the outer mitochondrial membrane [250]. In striated muscles, mitochondria are mostly positioned in the inter-myofibrillar spaces adjacent to triads, although contacts between mitochondria and the longitudinal SR have also been observed [251,252,253,254]. Association with the SR is mediated by the voltage-dependent anion channel 1 (VDAC) [255]. In cardiac muscle, it has been proposed that VDAC interacts with RyR2 [256]. Mitofusin-2 (MFN-2) also contributes to tethering SR and mitochondria in skeletal muscles [257,258]. Interestingly, during mouse skeletal muscle development, mitochondria redistribute from a longitudinal arrangement towards a triad-associated distribution, with an increase in the frequency of contacts until the first four months after birth [258]. These contacts are aligned with domains of the inner mitochondrial membrane enriched in the mitochondrial Ca^2+^ uniporter (MCU), a low-affinity Ca^2+^ transport protein (Figure 4). It has been shown that RyR1 opening results in the formation of local micro-domains, where Ca^2+^ concentration may be up to 10 folds higher than in the rest of the cytoplasm [259]. This high Ca^2+^ concentration allows the transfer of Ca^2+^ to the mitochondrial matrix through the uniporter [260]. From a functional point of view, increases in mitochondrial Ca^2+^ concentration regulate ATP production by enhancing the synthesis of NADH and FADH2 [261,262], thus linking ATP production to muscle demand. Accordingly, slow-twitch fibers in skeletal muscle that are highly oxidative have the highest mitochondrial content compared to fast-twitch fibers; as a result, MCU deletion results in impairment in muscle force and in oxidative metabolism [263]. In cardiac muscle, the MCU-mediated Ca^2+^ influx was shown to play an important role during the fight-or-flight response following catecholaminergic stimulation [264].

Alterations of SR/mitochondria interaction or of mitochondrial morphology and function in striated muscles are often observed in human muscle diseases or mouse models of muscle diseases characterized by alterations in Ca^2+^ signaling [251,265,266,267]. However, the exact mechanisms linking alterations of MAM, mitochondrial function, and muscle diseases are still to be defined.

## 7. The SR and the ER: Two Faces of a Single Organelle

This review started with the statement that the SR is a specialized form of ER of muscle cells dedicated to Ca^2+^ handling. One may therefore ask what happened to the ER in striated muscle cells. In eukaryotic cells, the ER is dedicated to Ca^2+^ storage, protein synthesis and folding, and lipid and sterol synthesis. Although ER-related functions are clearly present in striated muscles, the distribution of the ER within the SR membranes is less evident. The recognized idea is that the SR and the ER represent a continuous membrane system made of different specialized subdomains [268]. Compartmentalization of ER and SR markers was demonstrated in skeletal muscle fibers, where ER-specific proteins were detected at the perinuclear region and in two distinct rough ER sub-compartments: the first, in correspondence of the I band that does not apparently contain ER exit sites, and the second, near the Z disk, which sustains export activity towards the Golgi [106,269]. Similarly, in cardiac muscle cells, protein synthesis was found to occur not only around the nuclei but also within ER/SR membranes surrounding the sarcomeres; interestingly, translation of some SR membrane proteins has been proposed to occur starting from pools of mRNAs that are transported from the perinuclear region towards ER/SR protein-synthesis sites through a microtubule-dependent system [270]. A less-clear distinction can be observed in smooth muscle, where ER and SR markers are basically overlapping [271].

### Additional Contact Sites Contributed by the ER/SR

In eukaryotic cells, the ER is engaged in a complex network of interactions with many intracellular components, such as the nuclear envelope (NE), the plasma membrane, mitochondria, the Golgi complex, peroxisomes, endosomes, lysosomes, and lipid droplets [272]. All these interactions have been less characterized in striated muscle cells. Lysosome–SR nanojunctions were described in pulmonary arterial myocytes (Figure 4). These junctions were identified between clusters of lysosomes and perinuclear regions of the SR rich in RyR3 and are believed to provide an intracellular structure involved in the regulation of specific Ca^2+^-signaling events. These junctions were identified based on the observation that nicotinic acid adenine dinucleotide phosphate (NAADP), a Ca^2+^-mobilizing second messenger, induced the release of Ca^2+^ from a lysosome-related Ca^2+^ store, and this was amplified by the generation of intracellular Ca^2+^ waves supported by RyR3 opening, following a CICR mechanism [273]. According to these studies, lysosomes act as a trigger zone through which Ca^2+^ is released with a quantal pattern; upon reaching an activation threshold, RyR3 channels open, and calcium oscillations are generated. More recently, nanojunctions between lysosomes, SR, and mitochondria have been described at the ultrastructural level in rabbit ventricular myocytes. At difference with junctions observed in pulmonary arterial myocytes, in cardiac myocytes, lysosomes are distributed with a periodicity consistent with the length of a sarcomere, allowing the association with specific areas of the SR [274]. The role of NAADP-mediated signaling in cardiac muscle cells is suggested to be involved in intracellular Ca^2+^ regulation, mainly following beta-adrenergic stimulation [275].

In striated muscle cells, stacks of ER/SR cisternae distribute in the perinuclear regions and connect with the nuclear envelope (NE) with functions correlated with protein synthesis and the regulation of intra-nuclear Ca^2+^ signaling (Figure 4). The NE is formed by two phospholipid bilayers that form the inner nuclear membrane and the outer nuclear membrane, outlining an internal space called the perinuclear cisternae. The outer nuclear membrane is continuous with the membrane of the rough ER; this connection results in the formation of common luminal space with the ER, which, in muscle cells, allows the arrangement of a continuous Ca^2+^ storage system between the SR and the NE [276]. EM analysis showed the existence of membrane invaginations of the inner nuclear membrane that enter the nuclear matrix to support different functions, including intranuclear Ca^2+^ signaling [277]. InsP_3_R, RyRs, and SERCA have been described at these sites; in striated muscle cells, Ca^2+^ release is apparently mainly regulated by RyRs [278,279], while both InsP_3_R and RyRs are present in smooth muscle cells [280]. In cardiac myocytes, STIM1 and Orai1 have also been described in the inner nuclear membrane, suggesting that the intranuclear Ca^2+^ signaling may be more complex than previously expected [281].

An additional issue is represented by the SR/nuclei juxtaposition within the overall skeletal muscle architecture. In skeletal muscle, nuclei are positioned at the fiber periphery between myofibrils and the sarcolemma, in proximity to T-tubules. The positioning of nuclei is regulated by different mechanisms, including the formation of a specific microtubule network, and is critical for muscle function; indeed, improper myonuclear localization is observed in muscle diseases, such as centronuclear myopathy and muscular dystrophies [282]. The molecular link supporting nuclear juxtaposition with the SR is currently not known. A recent work performed in skeletal muscles of *Drosophila melanogaster* suggests that this interaction may be mediated by a three-partner connection where amphiphysin 2 on TT binds, at the same time, Ma2/d, a protein present on both TT and SR membranes, and, at the nuclear membrane, Msp30, a protein previously described to mediate connections between the NE and the Z disks [283,284,285].

## 8. Concluding Remarks

This review provides an updated view of the structural, molecular, and functional organization of the SR in skeletal muscle, highlighting some of the specialized domains whereby this organelle supports activation of muscle contraction. The historical knowledge that the terminal cisternae of the SR engage with the T-tubules in forming the triad, which represents the first identified membrane contact site, is now expanded by evidence that in skeletal muscle, the SR participates in the assembly of at least two more types of membrane-contact sites.

In one case, the identification of CEUs, dynamic membrane contact sites consisting of stacks of SR cisternae and the extension of T tubules, structurally and functionally different from triads, provides evidence of how skeletal muscle fibers can sustain refilling of intracellular Ca^2+^ stores under conditions of prolonged activity. The second contact site, MAM, between the SR and mitochondria, has the functional role of synchronizing the mitochondrial energy production rate with the metabolic demand of muscle fibers. Altogether, these new findings are changing the traditional view of the SR as a static organelle by depicting an unexpected dynamic ability to extend its complex structure to support muscle function.

At the same time, there are several points that remain to be clarified. A major open question relates to the precise positioning of triads at the boundary of the I and A bands of the sarcomere. This highly precise pattern, evident in EM micrographs for about a half-century, is waiting for the identification of the proteins that tether triads to the sarcomere with such a regularly repetitive precision. Concerning CEU, we need to identify the molecules that support the dynamic assembly and disassembly of CEU and how Orai1 and STIM1 are recruited at these sites. Mitochondria, like many other organelles, in skeletal muscle, are localized with a well-defined pattern with respect to triads and sarcolemma, a pattern that differs in slow- and fast-twitch fibers. Additionally, in this case, the molecules that anchor mitochondria to these domains of muscle fibers remain to be identified.

In conclusion, we can anticipate that the above list is, certainly, still missing additional SR domains that are yet to be identified; these might probably involve activities related to the regulation of Ca^2+^ homeostasis, but more likely, additional functions necessary to keep skeletal muscle fibers healthy and fully functional.

## Figures and Tables

**Figure 1 biomolecules-12-00488-f001:**
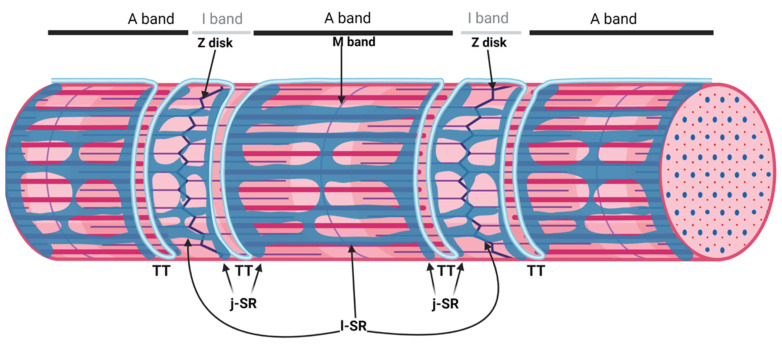
Organization of the sarcoplasmic reticulum in skeletal muscle cells. The regular alternation of anisotropic (A) and isotropic (I) bands along a single myofibril is depicted in the upper part of the image. A and I bands are bisected by the M band and the Z disk, respectively. The portion of the myofibril between two Z disks is the sarcomere. The SR is composed of tubules and cisternae surrounding each myofibril. The elongated tubules are known as longitudinal SR (l-SR); they are dedicated to the removal of Ca^2+^ from the cytosol and are localized around the A and I band of each sarcomere. At the borders between the A and I band, the l-SR merges to form the terminal cisternae. These are positioned at the opposite sides of a transverse tubule (TT); the structure formed by two terminal cisternae and one TT is called a "triad". The region of the terminal cisternae that opposes the TT membrane is called junctional SR (j-SR). Adapted from “Myofibril Structure” by BioRender.com (2022). Retrieved from https://app.biorender.com/biorender-templates, accessed on 14 March 2022.

**Figure 2 biomolecules-12-00488-f002:**
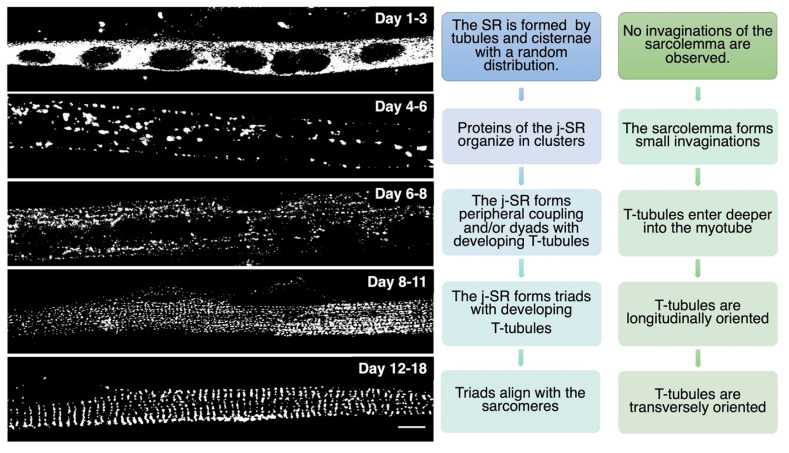
In vitro differentiation of primary rat myocytes.Primary rat myocytes induced to differentiate for 1 to 18 days were decorated with antibodies against RyR1 to label the j-SR. At the beginning of differentiation, RyRs show a diffuse distribution in the SR. Starting from day 4 of differentiation, RyRs form clusters, and the SR progressively forms peripheral couplings and/or diads with the TT. Triads form during the following days, and at the end of differentiation, they acquire their transverse orientation and localize at the borders between the A and I bands of the sarcomeres.

**Figure 3 biomolecules-12-00488-f003:**
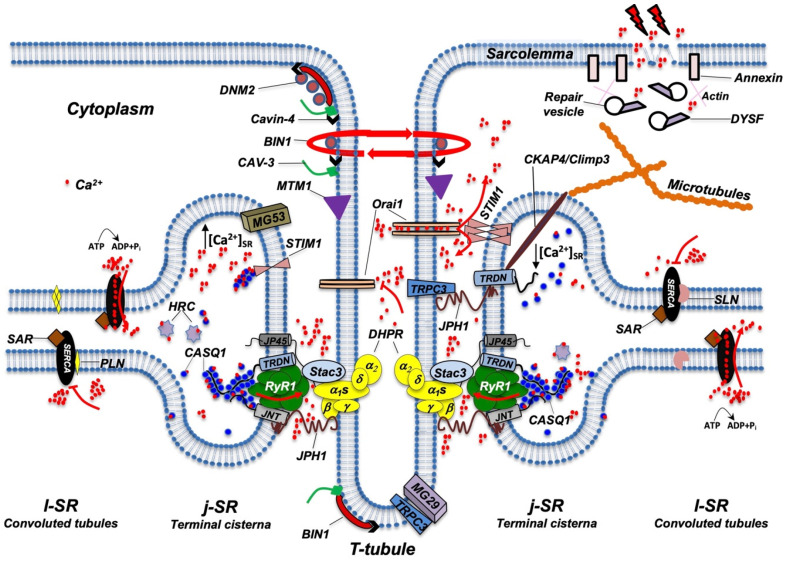
Schematic representation of the main proteins accommodated in TT, j-SR, and l-SR. Protein localization and reciprocal interactions are schematized as detailed in the text. Red arrows indicate Ca^2+^ fluxes (red dots) through RyR1, Orai1, and SERCA pumps. RyR1 opens following interaction with DHPR; Orai1 opens following interaction with STIM1 aggregates, which in turn are induced by a reduction in Ca^2+^ levels in the SR; SERCA pumps actively transport Ca^2+^ from the cytosol to l-SR; PLN or SLN act as SERCA inhibitors. DNM2, Cavin-4, BIN1, CAV-3, and MTM1 are involved in the maintenance of TT architecture and stability. They also participate in TT formation (not shown) and, together with DYSF, contribute to vesicle trafficking during the repair of the damaged plasma membrane (see text for additional details). For simplicity, not all proteins and/or protein complexes depicted, including cytoskeleton components, are positioned on both sides of the triad, as it occurs physiologically. The following is a list of acronyms depicted in Figure 3: BIN1 (Bridging integrator-1/Amphiphysin 2); CASQ1 (Calsequestrin 1); CAV-3 (Caveolin-3); CKAP4 (Cytoskeleton-associated protein 4/Climp63); DHPR (dihydropyridine receptor); DNM2 (Dynamin 2); DYSF (Dysferlin); HRC (Histidine-Rich Calcium binding protein); JNT (Junctin); JP45 (J-SR protein 1); JPH1 (Junctophilin 1); j-SR (junctional sarcoplasmic reticulum); l-SR (longitudinal sarcoplasmic reticulum); MG29 (Mitsugumin-29); MG53 (Mitsugumin-53); MTM1 (Myotubularin); PLN (Phospholamban); RyR1 (Type 1 Ryanodine Receptor); SAR (Sarcalumenin); SERCA (Sarco/Endoplasmic Reticulum Calcium ATPase); SLN (Sarcolipin); STIM1 (Stromal Interaction Molecule 1); TRDN (Triadin); TRPC3 (Transient Receptor Potential Cation Channel 3); T-tubule (transverse tubule). Adapted from [23].

**Figure 4 biomolecules-12-00488-f004:**
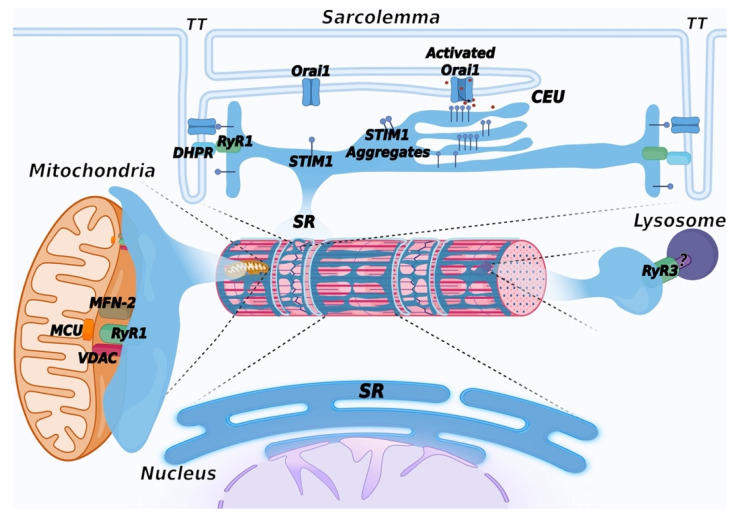
Schematic representation of membrane contact sites contributed by the SR. In addition to triads, the SR contributes to the formation of additional membrane contact sites in muscle cells. Depletion of intracellular Ca^2+^ stores in the SR induces activation of SOCE, mediated by a physical interaction between STIM1, a Ca^2+^ sensor of the SR and Orai1, a Ca^2+^ channel located in TT, allowing entry of Ca^2+^ from the extracellular space. Repetitive stimulation of muscle contraction was found to promote SR and TT remodeling to form additional sites of interaction between STIM1 and Orai1, called calcium entry units (CEU). These are formed by stacks of flat cisternae of the SR that make contact with elongated tubules extending from TT. In striated muscles, mitochondria are mostly positioned adjacent to triads. Association with the SR is mediated by the voltage-dependent anion channel 1 (VDAC) and RyR. Mitofusin-2 (MFN-2) also contributes to tethering SR and mitochondria. These contact sites are aligned with the inner mitochondrial membrane, where the mitochondrial Ca^2+^ uniporter (MCU) is located. Lysosome–SR nanojunctions mediated by RyR3 channels have been first described in pulmonary arterial myocytes; the RyR3 interactor present on the lysosomal membrane is not known and is therefore indicated with a question mark (?). More recently, these junctions have also been observed in cardiac muscle, while no data are currently available concerning skeletal muscle. The outer nuclear membrane is continuous with the membrane of the SR, allowing the arrangement of a continuous Ca^2+^ storage system between the SR and the nuclear envelope. The inner nuclear membrane forms invaginations that enter the nuclear matrix to support intra-nuclear Ca^2+^ signaling. Created with BioRender.com, accessed on 14 March 2022.

**Table 1 biomolecules-12-00488-t001:** Major subtypes of RyR1-related congenital myopathies.

	Causative Gene (S) *	Inheritance	Histological Features	Clinical Features
**Central Core Disease (CCD)**	*RyR1* > 90% *CACNA1S*	AD or AR	✓ Centrally located, well-demarcated cores, spanning the whole fiber axis Predominance in type 1 fibers Increased central nuclei [133]	✓ Infantile non-progressive hypotonia and motor development delay Mild proximal muscle weakness Respiratory distress High arched palate Craniofacial dysmorphism
**Multiminicore Disease (MMD)**	*RyR1* *CACNA1S*	AR	✓ Numerous cores in a limited area on longitudinal section Multiple internally located nuclei Predominance in type 1 fibers [133,134,135,136]	✓ Axial muscle weakness, scoliosis, respiratory insufficiency, and limb joint hyperlaxity Ophthalmoplegia Arthrogryposis Hand amyotrophy
**Centronuclear Myopathy (CNM)**	*RyR1*~12%	AR	✓ Centralized and internalized nuclei Peripheral halos depleted of oxidative activity Cores [133,134,135,136]	✓ Non-progressive proximal muscle weakness Non-progressive hypotonia
**Congenital Fibre Type Disproportion (CFTD)**	*RyR1*~20%	AR	✓ Fiber size disproportion (35–40% of type 1 fibers are smaller in size than type 2 fibers) Age-related development of rods, cores, and central nuclei [133,134,135,136]	✓ Static or slowly progressive muscle weakness Respiratory and proximal axial weakness Ophthalmoplegia Dysphagia Facial muscle weakness
**Dusty Core Disease (DUCD)**	*RyR1*	AR	✓ Irregularly sized/shaped “Dusty” cores (reddish-purple granular material deposition) spanning 10 to 50 sarcomeres Myofibrillar disorganization [133,134,135,136]	✓ Ocular involvement (eyelid ptosis, ophthalmoplegia)
**Core Rod Myopathy (CRM)**	*RyR1*	AD or AR	✓ Nemaline bodies (rods), clustered or widely distributed along the fibers Central cores [133,134,135,136]	✓ Non-specific clinical features, including: hypotonia, muscle weakness, scoliosis, and respiratory insufficiency
**Malignant Hyperthermia (MH)**	*RyR1* *CACNA1S* *STAC3*	AD	✓ No histological features can be found in muscle fibers from MH patients [119]	✓ Uncontrolled contractures and muscle rigidity Hyperthermia Hyperkalemia Hypermetabolism Cardiac arrhythmia

***** Mutations in additional genes are known to be causative for the listed congenital myopathies. See Refs. [119,133,134,135,136]. Autosomal-dominant (AD), autosomal-recessive (AR).

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
