# Peer review of "The Sarcoplasmic Reticulum of Skeletal Muscle Cells: A Labyrinth of Membrane Contact Sites"

_biomolecules, 2022, doi:10.3390/biom12040488_

Round 1
Reviewer 1 Report
Comments to « The sarcoplasmic reticulum of skeletal muscle cells: a labyrinth of membrane contact sites” by Rossi et al.
Skeletal muscle contraction is a complex biological phenomenon in which specialized biochemical interactions take place in structurally highly specialized intracellular structures, maintained by a complex set of dedicated protein components within the myotube. Specialized membrane sub-compartments of the plasma membrane and the sarcoplasmic reticulum play essential roles by permitting the ordered fluxes of calcium ions into and from the sarcoplasm, required for contraction and relaxation. These highly organized structures are established during rhabdomyocyte differentiation, and is maintained by non-covalent interactions of several specialized proteins, the defects of which, mostly due to genetic anomalies, can lead to various forms of myopathies, with associated characteristic microscopic structural, as well as functional defects.
In this paper the molecular mechanisms responsible for the establishment and the maintenance of the specialized sarcoplasmic reticulum and plasma membrane structures of striated muscle is summarized, and the function of these membranes, as well as that of the key proteins of excitation-contraction coupling and of relaxation are discussed. This paper is timely, well written, easy to read, and will interest a wide range of readers working in muscle physiology or the molecular pathology of myopathies. In addition, the description of the molecular organizing principles that allow the establishment and functioning of complex and specialized intracellular membrane structures involved in calcium homeostasis, fluxes and signaling may interest also readers outside of the field of muscle biology, as similar mechanisms may also be active in specialized intracellular calcium compartments in other cell types such as, for example, various types of neurons or platelets. The laboratory of the Authors is recognized internationally for their work on muscle molecular and developmental biology, as well as muscle molecular pathology.
Comments:
The understanding of the structural/spatial organization of myotube membrane structures is essential for the understanding of the paper. This is adequately discussed in the text. The Reviewer believes, however, that this should be supported by several Figures. The overall structure of the membranes, as well as their spatial relation with the contractile apparatus should be depicted and annotated regarding a fully differentiated myotube. The illustration of the various key steps of rhabdomyocyte differentiation, in terms of the complexification of membrane structures as discussed in the Paper, would be very useful as well.
In addition, several other Figures depicting the interactions discussed in the paper, of specific membrane proteins and lipids involved in the generation of membrane curvature, invagination/tube formation and the establishment and stabilization of plasma membrane-sarcoplasmic reticulum and mitochondrial interactions and contact sites should be included. Moreover, Figures on ECC, allosteric RyR/DHPR interactions and SOCE, should be illustrated as well. The Reviewer believes that his would greatly enhance the clarity of the Paper for a wide range of readers.
In addition, because Authors discuss various monogenic myopathies, it would be interesting to include electron-microscopical images showing characteristic membrane morphological changes if available, or as a minimum, specific References should be given to such morphological works. Also, it would be interesting to discuss, whether the molecular basis of centronuclear localization in some myopathies has been established.
Skeletal and cardiac muscle are discussed in detail in this work. Maybe smooth muscle and, eventually other simpler contractile cells could also be briefly mentioned or shown for morphological comparison, eventually. This would nicely illustrate the progressive complexification of the membrane structures along the process of muscle specialization.
Minor:
Please use boldface or plain text consistently when various proteins are named. In the Manuscript received by the Reviewer lanes are not numbered, unfortunately.
Abstract : “concerning the sarcoplasmic reticulum biology…” : concerning the biology of sarcoplasmic reticulum…
Introduction : “dedicated to calcium ions (Ca2+) handling…” : dedicated to calcium ion (Ca2+) handling…
“theRyanodine…” : the Ryanodine
“l-SR tubules volume…” : the volume of l-SR… or: l-SR tubule volume
“The triad is the membrane structure that support ECC…” : that supports…
“achieving its definitive localization at around three weeks of age…” : in mice…
Please explain “dyspedic” and “dysgenic”
Amphiphysin2 or Amphiphysin-2 ?
“is a member of a family of ubiquitary proteins…” : ubiquitous proteins…
“DYSF, together with the membrane-associated actin binding protein Annexin, is concentrated in the vicinity of membrane injury…” : which annexin ?
“since, in addition to recruit annexins to the injury sites…” : since, in addition to recruiting annexins to the injury sites
“during membrane damage repair, it is recruited at damaged membrane sites by Polymerase I and transcript release factor…” : the identity of polymerase I and transcript release factor is not clear.
“Surprisingly, any MG53 mutation associated to the onset of skeletal muscle pathologies have been described so far.” : Surprisingly, no MG53 mutation associated to the onset of skeletal muscle pathologies has been described so far.
“In vitro studies on mouse primary skeletal myotubes also showed that MG29 interacts with the canonical-type transient receptor potential cation channel 3 (TRPC3)…” : please specify the cellular location of this interaction.
“MTM1 dephosphorylates Phosphatidyl-Inositol-3-P (PtdIns3P) and Phosphatidyl-Inositol-3,5-P… : …3,5-bisohosphate.
Please use “phosphatidyl…” and “inositol…”
“The α-helical region is highly conserved, and provide mechanical elasticity…” : provides
“presenting triads alteration and impaired ECC…” : presenting triad alterations… or : presenting alterations of triads…
“In the heart, JPH2 mis-localization or downregulation correlates with remodeling and loss of TT… : this error of localization in the heart is due to what ?
“In addition to its structural role, JPHs can also bind…” : In addition to their structural role, JPHs can…
“Recently, different studies showed that…” : “various studies” or : “several studies”
“both JPH1 and JPH2 are targets of the calcium-depended protease calpain…” : calpains are a multigene family; it would be nice if Authors could specify which calpain is involved.
“JPH degradation results in modification in triads and diads geometry…” : JPH degradation results in the modification of the geometry of triads and diads…
“At variance with RyR1 channels, RyR3 channels are activated by a Ca2+-induced Ca2+-release mechanism… : it would be useful to discuss the difference between CICR and direct DHPR/RyR allosteric coupling here for clarity. “at variance” is not sufficiently specific here, because this issue is discussed only later in the text.
“that represent binding site for several auxiliary proteins…” : “that represents a binding site for several auxiliary proteins…” or : “that represent binding sites for several auxiliary proteins…”
“Additional physiological or pharmacological RyR1 regulators include Mg2+, Homer 1c, the scorpion venom imperatoxin, halothane, dantrolene and PCB95 (111, 117, 118). Finally, RyRs can be also regulated by post-translational modification, such as oxidation, nitrosylation, phosphorylation/dephosphorylation cycles that occur via several kinases, like PKA, PKG, Ca2+/CaM-dependent protein kinase II (CaMKII) and phosphatases (PP1, PP2A, and PDE4D3) (111, 119).”
The Reviewer believes that it would be appropriate to mention ryanodine, ruthenium red, methylxanthines and 4-chloro-m-cresol, as well. The brief mention of the role of dantrolene as a useful agent in malignant hyperthermia would also be appropriate here, or in the myopathies section.
“each composed by 6 alpha helices…” : composed of… ?
“RyR1 is also able to regulate DHPR activity by the so called ‘retrograde’ coupling effect…” : please briefly discuss the nature and the functional consequences of this retrograde effect.
“It is an adaptor protein that supports trafficking of the a1s subunit and regulate coupling of DHPR with
RyR1…” : please specify here that a1s refers to a DHPR subunit.
Table 1. needs extensive improvement by adequate formatting (using, for example, spaces, alignment, separating lines or different colors for lanes or sections).
“Peripheral halos depleted of oxidative activity” meaning in muscle cross-sections ?
“…occurring following exposure to succinylcholine and volatile anesthetics…” : “and” or “or” ?
In the understanding of the Reviewer, volatile anesthetics alone are already capable to induce malignant hyperthermia.
Please italicize “versus”, “via” etc.
“and Trisk 49, which do not localize…” : “which does not localize…” ?
Please replace “Ca2+” by “Ca2+” (“2+” uppercase) in the text.
“In skeletal muscle, two isoforms of CASQ have been identified…” : it would be appropriate to specify here that CASQ means calsequestrin. Please state explicitly that calsequestrin is an ER intra-luminal soluble protein in this system.
“when the ionic strength increases, these domains fold together…” : Authors probably mean increases of ER luminal calcium concentrations.
“CASQ1 polymerization is considered as a reversible process…” : “CASQ1 polymerization is considered a reversible process…” ?
“depol-ymerization” : “de-polymerization” ?
“…or even have no effect on RyRs opening…” : “…or even have no effect on the opening of RyRs…” or : “…or even have no effect on RyR opening…”
“They exhibit significantly reduced total Ca2+ content…” meaning in the sarcoplasmic reticulum ?
“Indeed, in CASQ1 knockout mice, SOCE is constitutively active, likely to compensate for the reduction in total releasable SR Ca2+ content…” : this requires some discussion, given that STIM is considered to sense free, rather than calsequestrin-bound calcium. Is enhanced SOCE related to shifts of the dynamical equilibrium of the dissociation of calcium-protein complexes and to changes in calcium transport ?
“It is composed of a conserved N-terminal domain, a C-terminal cysteine-rich region and a central histidine-rich repeat region which allow HRC to bind Ca2+ and to interact with triadin…” : “allow” or “allows” ?
“Although considered as secondary calcium-binding protein…” : “as a secondary…”
“HRC appears to play a not-secondary role…” : a non-secondary role ?
“SERCAs belong to the P-type ATPases family, a large superfamily of integral membrane proteins that pump ions and lipids across cellular membranes…” : please include examples and References for lipid transport by P-type ATPases.
“In vertebrates, SERCA pumps are encoded by three different genes: SERCA1-3 or ATP2A1-3.” Please clarify in the text that SERCA is the name of the proteins, whereas ATP2A1-3 denote the corresponding genes.
“alternative splicing occurring in the C-terminus of the main transcript…” : please note, that if alternative splicing is discussed in the context of more than one gene, than the use of “main transcripts” is more appropriate.
Ibid.: the Reviewer is not convinced that Ref 194 alone (Brandl et al. J.Biol.Chem., 1984) alone covers the splicing patterns of all SERCA transcripts, in particular, regarding SERCA2 and SERCA3.
Also, please note, that Ref 197 is misplaced; this Ref. should be moved to the text dealing with SERCA2 alternative splicing (where Ref. 198 is located).
“SERCA2b is expressed in all cell types…” : please note that erythrocytes express no detectable SERCA2b.
“SERCA1a displays a maximal activity that is two folds higher…” : two-fold higher ? Also, please note that significant differences exist regarding the calcium affinity of SERCA2b versus SERCA3.
“Additional contact sites participated by the SR…” : contributed by ?
“requires a coordinated interactions…” : requires a coordinated interaction…
“bothregulating” : both regulating
Please note that the SOCE channels are usually spelled as “Orai”
The Reviewer believes that in Paragraph 5.1. it would be appropriate to briefly discuss PMCA-type calcium pumps and sodium-calcium exchangers, as these play a rather important role in the context of trans-plasma membrane calcium ion transport, and because calcium extrusion from muscle cells across the plasma membrane is closely related to the main theme of this paper.
Please add References for STIM/Orai mutation-related skeletal muscle phenotypes and plasma membrane-SR junction formation following intense prolonged muscle activity.
“(VDAC) (229)” : Authors are usually advised to avoid using adjacent brackets. “;” could be used instead.
“allows transfer of Ca2+ to the mitochondria matrix…” : “to the mitochondrial matrix” or “to the matrix of mitochondria” (Idem for “mitochondrial morphology” and “mitochondrial function”.)
“Accordingly, the contribution of the terminal cisternae and longitudinal tubules in sustaining the mechanisms of Ca2+ release and re-uptake in the context of the ECC mechanism is reported, along with a list of the principal proteins that participate in these processes. In addition, two more recently identified membrane contact sites formed by the SR are presented.” : This text in the “Concluding remarks” would be, in the opinion of the Reviewer, better suited to an Abstract-type section, and conveys little message in terms of a conclusion. Maybe Authors could consider replacing it by a text summarizing the great general biological complexity of skeletal muscle membrane morphogenesis and structural and functional specialization or something similar.
“extends our knowledge on how muscles can synchronize the energy production rate with the metabolic demand.” : “…their energy production rate with their metabolic demand” ?
“by depicting a complex network of in intracellular membrane contact sites.” : please delete “in”.
In "References" please use boldface consistently for publication year (see for example Refs. 1 and 54). Also, please adjust font for Ref. 241.
The Reviewer also would like to draw the attention of the Authors to the issue of consistent spelling of “Ca2+”. The Reviewer believes that whereas “Ca2+” and “Ca(2+) are correct spellings, “Ca2+” should be replaced by “Ca2+” (“2+” superscript) in the Manuscript and the References.
Author Response
Please see the attachement

Reviewer 2 Report
Rossi et al. discussed about the sarcoplasmic reticulum of skeletal muscle cells and the contact sites with the other organelles as indicated in the title. However, they only discussed in a short paragraph the contact sites between mitochondria and the ER/SR. What about the other organelles? Despite that, the review is well-written and present the last findings in the literature. An outlook and future directions for research in this field are necessary at the end of the review. Moreover, the lack of figures or pictures renders this review difficult to read due to the abundance of informations. At this stage, I don't recommend the publication of this review. However, if the authors provide a point-to-point responses to my questions, I could reconsider my decision.
Round 2
Reviewer 1 Report
The Authors addressed the issues raised by the Reviewer successfully, and the clarity of the Paper has been enhanced. The modifications involved the addition of several paragraphs of new text in the revised manuscript. The comments of the Reviewer are now related to this material.
Comments:
Fig. 2 : Maybe including a bibliographical reference in the Legend of this Figure would be appropriate (if the microscopic images shown are not original work.)
If Fig. 3 is not original art, a Reference could be added here as well. Please insert a line break at the beginning of the list of abbreviations (starting with BIN1) in the Legend.
“RyR1 and DHPR, although essential for ECC, have been shown to be dispensable for the formation of triads, since triads form in mouse models lacking either RyR1 (dyspedic mice), DHPR (dysgenic also called “dyspedic” mice since RyRs were defined as “junctional feet”), DHPR (also called “dysgenic” mice due to a natural mutation, the muscular dysgenesis in DHPR, that causes lethal paralysis in mice) or both proteins (19-21).” : Please reformulate this sentence, maybe cut it into several separate phrases, for syntax and clarity.
“In vitro studies on mouse primary skeletal myotubes also showed that MG29 interacts with the canonical-type transient receptor potential cation channel 3 (TRPC3), localized on the TT, suggesting the existence of an additional mechanism able to regulate Ca2+ transients during skeletal muscle contraction.” : localized “in” or “at” ?
“…a mechanism defined as to depolarization-induced-Ca2+-release…” : a mechanism defined as
depolarization-induced-Ca2+-release
“RyR2 channels are mainly expressed in cardiac muscle, where they are involved in cardiac ECC” : readers may be curious here whether this RyR isoform operates through a RyR1- or RyR3-type opening mechanism (conformational coupling or CICR).
“led to a significant decrease in mortality from more than 90% to less than 5%” : “led to a significant decrease in mortality, from more than 90% to less than 5%” or: “led to a significant decrease in mortality, i.e.: from more than 90% to less than 5%” or: “led to a significant decrease in mortality (from more than 90% to less than 5%).
“RyR2 was found non to be able to restore either orthograde or retrograde signaling” …found unable to… or : …found not to be able to…
“Autosomic dominant (AD), autosomic recessive (AR).” : …autosomal ?
“STIM1/ORAI interaction (184, 185).” : Orai
“such as Inositol 1,4,5-trisphosphate receptors…” : …inositol…
Please note that as per its title, Ref. 209 is about SERCA1 and 2a comparisons, not 2b and 3. Maybe citing work, for example, from the Wuytack or the Lytton laboratory would be adequate for SERCA2b and 3 comparative biochemistry.
“More than 10 protein variants are generated through alternative splicing occurring in the C-terminus of the main transcript” : please note that C- or N-terminus refers to protein sequences; transcripts have 5’- or 3’ termini.
“SERCA2b is expressed in almost all cell types…” : this is somewhat confusing, because “almost” is not defined clearly here. Readers may be curious, which cell type(s) are(is) negative.
“They can transport one Ca2+ for each hydrolyzed ATP and have a high affinity for Ca2+ when complexed with calmodulin…” : this phrase is somewhat confusing in its present form (“…Ca2+ when complexed with calmodulin…”); a clear distinction should be made between the notion of calmodulin-calcium complex-enhanced calcium affinity of PMCA enzymes (regarding the transported calcium ion) and the affinity of calcium to calmodulin (and the affinity of the calcium-calmodulin complex to PMCA.)
“where it represents the major mechanism of Ca2+extrusion from the cytoplasm” : Ca2+ extrusion…
“stacks of flat cisternae of the SR that take contact with elongated tubules…” : …that make contact…?
“In eukaryotic cells, the ER is dedicated to protein synthesis and folding, lipid and sterol synthesis.” Please note that the ER is also a very important intracellular calcium storage organelle in non-muscle cells. Calcium mobilization from the ER is central for SOCE and calcium signalling, and contains STIM1, IP3R etc.
“where ER and SR markers are basically overlapped” : “are overlapping” or “markers basically overlap”
“the ER is engaged in a complex network of interaction with many intracellular components…” : network of interactions…
“that the Nicotinic acid adenine dinucleotide phosphate…” : that nicotinic acid adenine dinucleotide phosphate…
“induced the release of Ca2+ form a lysosome-related Ca2+ store that was amplified by the generation of intracellular Ca2+ waves supported by RyR3 opening…” : “induced the release of Ca2+ form a lysosome-related Ca2+ store, and this was amplified by the generation of intracellular Ca2+ waves supported by RyR3 opening…” (or: “which was amplified…”)
“in in rabbit ventricular myocytes” : in rabbit ventricular myocytes
“Nuclei positioning is regulated by…” : The positioning of nuclei is regulated by… or: “Nucleus positioning is regulated by…”
“microtubule network and is critical for…” : microtubule network, and is critical for…
“participates in the assembly of at least two more membrane contact sites…” : participates in the assembly of at least two more types of membrane contact sites…
“has the functional role of synchronizing their energy production…” : "has the functional role of synchronizing mitochondrial energy production…"
“An open major question…” : A major open question…
“evident in EM micrographs for about half a century…” : ...since about a half century…
“CEU and how…” : CEU, and how…
“Also in this case, which molecules anchor mitochondria to these domains of muscle fibers are yet waiting for identification.” : "Also in this case, the molecules that anchor mitochondria to these domains of muscle fibers remain to be identified."
“In conclusion, we can anticipate that the above list is, certainly, still missing additional SR domains that are yet to be identified; these might probably include activities related to regulation of Ca2+ homeostasis, but more likely, additional functions necessary to maintain skeletal muscle fibers healthy and fully functional.” Please note that a domain is not an activity.
“In conclusion, we can anticipate that the above list is, certainly, still missing additional SR domains that are yet to be identified; including domains probably involved in activities related to regulation of Ca2+ homeostasis, as well as additional domains necessary to maintain skeletal muscle fibers healthy and fully functional.” or something similar. Or : "...these might probably involve activities..." or "...that may be related to activities involved in..."
Finally, please note that the Manuscript received by the Reviewer is a pdf copy of the new word text file with the modifications (insertions, deletions) highlighted. This is very useful for the reviewing process. However, it will be important to check the Paper before publication in its final form (all modifications accepted, deletions not shown; in “deactivate Track changes” mode) for eventual typographical errors at junctions of old and new text, as such eventual errors are difficult to discern in the Manuscript in its present form.
Reviewer 2 Report
I have no further comments.